# *LoopReg*: Self-supervised Learning of Implicit Surface Correspondences, Pose and Shape for 3D Human Mesh Registration

**Bharat Lal Bhatnagar**
Max Planck Institute for Informatics
Saarland University
bbhatnag@mpi-inf.mpg.de

**Cristian Sminchisescu**
Google Research
sminchisescu@google.com

**Christian Theobalt**
Max Planck Institute for Informatics
Saarland University
theobalt@mpi-inf.mpg.de

**Gerard Pons-Moll**
Max Planck Institute for Informatics
Saarland University
gpons@mpi-inf.mpg.de

## Abstract

We address the problem of fitting 3D human models to 3D scans of dressed humans. Classical methods optimize both the data-to-model correspondences and the human model parameters (pose and shape), but are reliable only when initialized close to the solution. Some methods initialize the optimization based on fully supervised correspondence predictors, which is not differentiable end-to-end, and can only process a single scan at a time. Our main contribution is *LoopReg*, an end-to-end learning framework to register a corpus of scans to a common 3D human model. The key idea is to create a self-supervised loop. A backward map, parameterized by a Neural Network, predicts the correspondence from every scan point to the surface of the human model. A forward map, parameterized by a human model, transforms the corresponding points back to the scan based on the model parameters (pose and shape), thus closing the loop. Formulating this closed loop is not straightforward because it is not trivial to force the output of the NN to be on the surface of the human model – outside this surface the human model is not even defined. To this end, we propose two key innovations. First, we define the canonical surface implicitly as the zero level set of a distance field in $\mathbb{R}^3$, which in contrast to more common UV parameterizations $\Omega \subset \mathbb{R}^2$, does not require cutting the surface, does not have discontinuities, and does not induce distortion. Second, we diffuse the human model to the 3D domain $\mathbb{R}^3$. This allows to map the NN predictions forward, even when they slightly deviate from the zero level set. Results demonstrate that we can train *LoopReg* mainly self-supervised – following a supervised warm-start, the model becomes increasingly more accurate as additional unlabelled raw scans are processed. Our code and pre-trained models can be downloaded for research [5].

## 1 Introduction

We propose a novel approach for model-based registration, i.e. fitting parametric model to 3D scans of articulated humans. Registration of scans is necessary to complete, edit and control geometry, and is often a precondition for building statistical 3D models from data [82, 48, 58, 11].
Classical model-based approaches optimize an objective function over scan-to-model correspondences and the parameters of a statistical human model, typically pose, shape and non-rigid displacement. When properly initialized, such approaches are effective and generalize well. However, when the variation in pose, shape and clothing is high, they are vulnerable to local minima.

To avoid convergence to local minima, researchers proposed to use predictors to either initialize the latent parameters of a human model [32], or the correspondences between data points and the model [60, 76]. Learning to predict global latent parameters of a human model directly from a point-cloud is difficult and such initializations to standard registration are not yet reliable. Instead, learning to predict correspondences to a 3D human model is more effective [10, 14].

Several important limitations are apparent with current approaches. First, supervising an initial regression model of correspondence requires labeled scans [60, 76, 81, 52], which are hard to obtain. Second, although some approaches use predicted correspondences to initialize a subsequent, classical optimization-based registration, this process involves non-differentiable steps.

What is lacking is a joint end-to-end differentiable objective over correspondences and human model parameters, which allows to train the correspondence predictor, self-supervised, given a corpus of unlabeled scans. This is our motivation in introducing *LoopReg*.

Given a point-cloud, a backward map, parameterized by a neural network, transforms every scan point to a corresponding point on the canonical surface (the human model in a canonical pose and shape). A forward map, parameterized by the SMPL human model [48], transforms canonical points under articulation, shape and non-rigid deformation, to fit the original point-cloud, see Fig. 1. *LoopReg* creates a differentiable loop which supports the self-supervised learning of correspondences, along with pose, shape and non-rigid deformation, following a short supervised warm-start.

The design of *LoopReg* requires several technical innovations. First, we need a continuous representation for canonical points, to be on the human surface manifold. We define the surface implicitly as the zero level set of a distance field in $\mathbb{R}^3$ instead of the more common approach of using a 2D UV parameterization $\Omega \subset \mathbb{R}^2$, which typically relies on manual interaction, and inevitably has distortion and boundary discontinuities [35]. We follow a Lagrangian formulation; during learning, NN predictions which deviate from the implicit surface are penalized softly. Furthermore, we interpret the 3D human model as a function on the surface manifold. We diffuse the function onto the 3D domain via a distance transform (Fig. 3), which allows to map the NN predictions forward, when they slightly deviate from the surface during learning.

In summary, our key contributions are:

- *LoopReg* is the first end-to-end learning process jointly defined over a parametric human model *and* the data (scan / point cloud) to model correspondences.

- We propose an alternative to classical UV parameterization for correspondences. We define the canonical human surface implicitly as the zero levelset of a distance field, and diffuse the SMPL function to the 3D domain. The formulation is continuous and differentiable.

- *LoopReg* supports self-supervision. We experimentally show that registration accuracy can be improved as more unlabeled data is added.

## 2   Related Work

In this section, we first broadly discuss existing work on correspondence prediction followed by approaches on non-rigid registration with special focus on methodology dealing with both.

**Correspondence prediction.**   Early 3D correspondence prediction methods relied on optical flow [16], parametric fitting [74], function mapping [7], and energy minimization [83, 50, 22, 21, 23]. Recent advancements include functional maps to transport real valued functions across surfaces [53, 52, 28, 31], learning from unsupervised data [65, 33, 65], implicit correspondences [71] and search for novel neural network architectures [20, 81, 79].

Aforementioned work focused primarily on establishing correspondences across isometric [54] or non-isometric shapes [69, 39, 80] but did not leverage such information for parametric model fitting.

**Model based registration.** In the context of articulated humans, classical ICP based alignment to parametric models such as SMPL [48] has been widely used for registering human body shapes[56, 18, 19, 58, 59, 34, 29] and even detailed 3D garments [57, 15]. Incorporating additional knowledge such as precomputed 3D joints, facial landmarks [42, 8] and part segmentation [14] significantly improves the registration quality but these pre-processing steps are prone to error at various steps. Though there exist approaches for correspondence-free registration [72, 70, 68], we focus on work that exploit correspondences for non-rigid registration [12]. Recent work have built on classical techniques such as functional maps [49], part-based reconstruction [27] and segmentation guided registration [44] and showed impressive advancements in the registration quality. Despite their

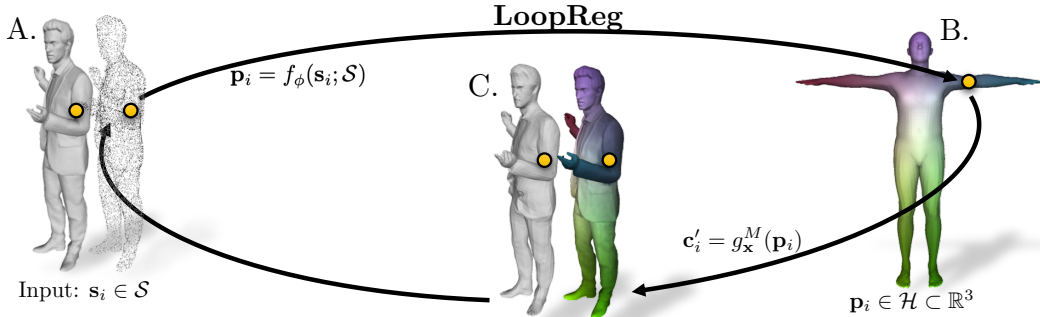

Figure 1: The input to our method is a scan or point cloud (A) $\mathcal{S}$. For each input point $\mathbf{s}_i$, our network CorrNet $f_\phi(\cdot)$ predicts a correspondence $\mathbf{p}_i$ to a canonical model in $\mathcal{H} \subset \mathbb{R}^3$ (B). We use these correspondences to jointly optimize the parametric model (C) and CorrNet under self-supervised training.

strengths, these approaches are not end-to-end differentiable wrt. correspondences. One of the major bottlenecks in making correspondences differentiable is defining a continuous representation for 3D surfaces.

**Representing 3D surfaces.** Surface parameterization is non-trivial because it is impossible to find a zero distortion bijective mapping between an arbitrary genus zero surface in $\mathbb{R}^3$ and a finite planar surface in $\mathbb{R}^2$. Prior work has tried to minimize custom objectives such as angle deformation [43], seam length together with surface distortion [45] and packing efficiency [47]. Attempts have been made to minimize global [37, 67] or patch-wise distortion using local surface features[85]. But the bottom line remains –a surface in $\mathbb{R}^3$ cannot be embedded in $\mathbb{R}^2$ without distortions and cuts. Instead, we define the surface as the zero levelset of a distance field in $\mathbb{R}^3$, and penalize deviations from it. Additionally, we diffuse surface functions to 3D, which allows us to predict correspondences directly in 3D. While implicit surfaces [55, 51, 24, 66, 25] and diffusion [38, 36] techniques have been used before, they have not been used in tandem to parameterize correspondences in model fitting.

**Joint optimization over correspondence and model parameters.** Prior work initialize correspondences with a learned regressor [60, 76, 61], and later optimize model parameters, but the process is not end-to-end differentiable. An energy over correspondence and model parameters can be minimized directly with non-linear optimization (LMICP [30]), which requires differentiating distance transforms for efficiency [9, 73]. In general, the distance transform needs to change with model parameters, which is hard and needs to be approximated [77] or learned [26]. A few works are differentiable both wrt. model parameters and correspondences [86, 75]; but their correspondence representation is only piece-wise continuous and not suitable for learning. Using UV maps, continuous and differentiable (except at the cut boundaries) correspondences can be learned [41, 40] jointly with model parameters, but inherent problems with UV parametrization still remain. Using mixed integer programming, sparse correspondences and alignment can be solved at global optimality [13]. In 3D-CODED [32] authors directly learn to deform the model to explain the input point cloud, but this limits the ability to make localized predictions, and the approach has not been shown to work with scans of dressed humans.
Our approach on the other hand is not only continuous and differentiable wrt. to both correspondences and model parameters, but also does not rely on the problematic UV parametrization.

## 3  Method

Current state of the art human model based registration such as [42, 8] require pre-computed 3D joints and keypoint/landmark detection. These approaches render the scans from multiple views, use OpenPose [1] or similar models for image based 2D joint detection, and lift the 2D joint detections to 3D. This is prone to error at multiple levels. Per-view joint detection may be inconsistent across views. Furthermore, when scans are point-clouds instead of meshes, they can not be rendered. Fig. 4 and 5 show that accurate scan registration is not possible for complex poses without this information. In *LoopReg*, we replace the pre-computed sparse joint information with continuous correspondences to a parametric human model. Our network *CorrNet* contains a backward map, that transforms scan points to corresponding points on the surface of a human model with canonical pose and shape. In a forward map, these corresponding points are deformed using the human model to fit the original scan, thereby creating a self-supervised loop. We start our description by reviewing the basic formulation

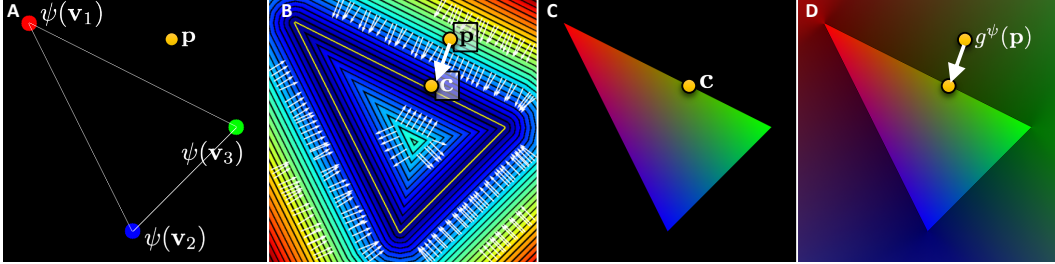

Figure 2: Illustration of the diffusion process. We diffuse the function $\psi(\cdot)$ (denoted as per-vertex colours), defined only on the vertices $(\mathbf{v}_1, \mathbf{v}_2, \mathbf{v}_3)$, to any point $\mathbf{p} \in \mathbb{R}^3$. Within the surface (in this example just a triangle), the function $\psi(\cdot)$ is diffused using barycentric interpolation (sub-figure C). For a point $\mathbf{p} \in \mathbb{R}^3$ beyond the surface, the function $\psi(\cdot)$ is diffused by evaluating the barycentric interpolation at the closest point $\mathbf{c}$ to $\mathbf{p}$, implemented by pre-computing a distance transform (sub-figure B). The result is a diffused function $g^{\psi}(\mathbf{p})$ defined, not only on vertices, but over all $\mathbb{R}^3$. Fig. 3 explains the same process for a complete human mesh.

of traditional model based fitting approaches, and follow with our self-supervised registration method. For improved readability we additionally tabulate all notation in the supp. mat.

### 3.1 Classical Model-Based Fitting

The classical way of fitting a 3D (human) model to a scan $\mathcal{S}$ is via minimization of an objective function. Let $M(\mathbf{v}_i, \mathbf{x}) : \mathcal{I} \times \mathcal{X}' \mapsto \mathbb{R}^3$, denote the human model which maps a 3D vertex $\mathbf{v}_i \in \mathcal{I}$, on the canonical human surface $\mathcal{M}_T \subset \mathbb{R}^3$ to a transformed 3D point after deforming according to model parameters $\mathbf{x} \in \mathcal{X}'$. For the SMPL+D model, which we use here, $\mathbf{x} = \{\boldsymbol{\theta}, \boldsymbol{\beta}, \mathbf{D}\}$ corresponds to pose $\boldsymbol{\theta}$, shape $\boldsymbol{\beta}$, and non-rigid deformation $\mathbf{D}$. The standard registration approach is to find a set of corresponding canonical model points $\mathcal{C} = \{\mathbf{c}_1, \ldots, \mathbf{c}_N\}$ (the *correspondences*) for the scan points $\{\mathbf{s}_1, ..., \mathbf{s}_N\}, \mathbf{s}_i \in \mathcal{S}$ and minimize a loss of the form:

$$L(\mathcal{C}, \mathbf{x}) = \sum_{\mathbf{s}_i \in \mathcal{S}} \mathrm{dist}(\mathbf{s}_i, M'(\mathbf{c}_i, \mathbf{x})), \tag{1}$$

where $\mathrm{dist}(\cdot, \cdot)$ is a distance metric in $\mathbb{R}^3$. Note that Eq. (1) uses continuous surface points $\mathbf{c}_i \in \mathcal{M}_T$, and $M'(\cdot)$ interpolates the model function $M(\cdot)$ defined for discrete model vertices $\mathbf{v}_i \in \mathcal{I}$ with barycentric interpolation.

Eq. 1 is minimized with non-linear ICP, which is a two step non-differentiable process. First, for every scan point a corresponding point on the human model is computed. Next, the model parameters are updated to minimize the distance between scan points and corresponding model points using gradient or Gauss-Newton optimizers. This alternating process is non-differentiable, which rules out end-to-end training. Our work is inspired by [75] which continuously optimizes the corresponding points and the model parameters. The trick is to parameterize the canonical surface with piece-wise mappings from a 2D space $\Omega$ to 3D $\mathbb{R}^3$ – per triangle mappings. This requires keeping track of the TriangleIDs and point location within the triangle when correspondences shift across triangles. Apart from being difficult to implement, this is not a suitable representation for learning correspondences as TriangleIDs do not live in a continuous space with metric. Furthermore, the optimization in [75] is instance specific.

### 3.2 Proposed Formulation

Instead of instance specific optimization, we want to automatize the model fitting process by leveraging a corpus of 3D human scans. Our key idea is to create a differentiable registration loop motivated by classical model-based fitting Eq. (1). We learn a continuous and differentiable mapping $f_{\phi}(\mathbf{s}; \mathcal{S}) : \mathbb{R}^3 \times \mathcal{S} \mapsto \mathcal{M}_T \subset \mathbb{R}^3$, with network parameters $\phi$, from the scan points $\mathbf{s} \in \mathcal{S}$ to the canonical surface (of the human model in canonical pose and shape) $\mathcal{M}_T$. Let $\{\mathcal{S}_j\}_{j=1}^{N_u}$ be a set of unlabeled scans, and $\mathcal{X} = \{\mathbf{x}_j\}_{j=1}^{N_u}$ be the set of unknown instance specific latent parameters per scan. The following self-supervised loss creates a loop between the scans and the model

$$L(\phi, \mathcal{X}) = \sum_{j=1}^{N_u} \sum_{\mathbf{s}_i \in \mathcal{S}_j} \mathrm{dist}(\mathbf{s}_i, M'(f_{\phi}(\mathbf{s}_i; \mathcal{S}_j), \mathbf{x}_j)), \tag{2}$$

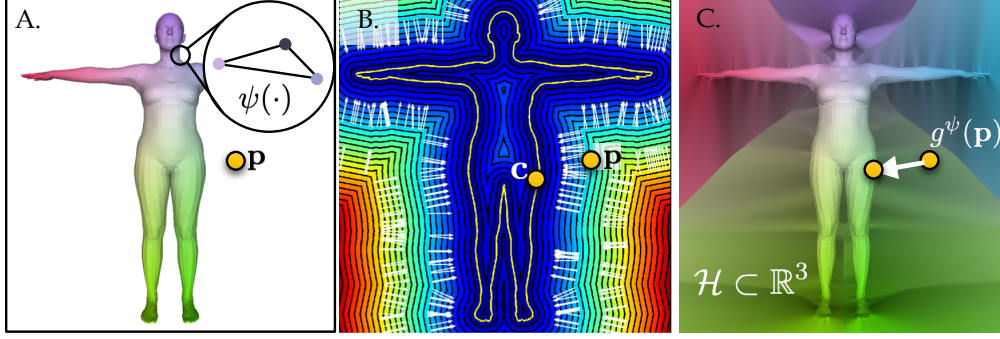

Figure 3: Illustration of the diffusion process on a human mesh. In sub-figure A, an arbitrary function $\psi(\cdot)$, defined over mesh vertices (illustrated as per vertex colors), is diffused to $\mathcal{H} \subset \mathbb{R}^3$ via a distance transform (sub-figure B). This results in a new function $g^\psi(\cdot)$ (sub-figure C).

where, in contrast to the instance specific Eq. (1), optimization in Eq. (2) is over a training set, and correspondences are predicted by the network, $\mathbf{c} = f_\phi(\mathbf{s}; \mathcal{S}_j)$. It is important to note that those correspondences network predicted, $\mathbf{c}$, need not be on the canonical surface. Outside the canonical surface, $M'(\cdot)$ is not even defined. The question then is how to minimize Eq. (2) end-to-end.

**Implicit surface representation.** To predict correspondences on $\mathcal{M}_T$, we need a continuous representation of its surface. One of our key ideas is to define the surface implicitly, as the zero levelset of a signed distance field. Let $d(\mathbf{p}) : \mathbb{R}^3 \mapsto \mathbb{R}$ be the distance field, defined as $d(\mathbf{p}) = \text{sign}(\mathbf{p}) \cdot \min_{\mathbf{c} \in \mathcal{M}_T} \|\mathbf{c} - \mathbf{p}\|$ taking a positive sign on the outside and negative otherwise. The surface is defined implicitly as $\mathcal{M}_T = \{\mathbf{p} \in \mathbb{R}^3 \mid d(\mathbf{p}) = 0\}$. But how can we satisfy the constraint $d(\mathbf{p}) = 0$ during learning? And how to handle network predictions that overshoot the surface?

**Diffusing the human model function to $\mathbb{R}^3$.** Imposing the hard constraint $d(\mathbf{p}) = 0$ during learning is not feasible. Hence, our second key idea is to diffuse the human body model to the full 3D domain (see Fig. 3, and Fig. 2 to better understand the diffusion process with a simple example). Without loss of generality, we use SMPL [48] as our human model throughout the paper, but the ideas apply generally, to other 3D statistical surface models [82, 64, 56, 17, 46].
The SMPL model applies a series of linear mappings to each vertex $\mathbf{v}_i \in \mathcal{I}$ of a template, followed by skinning. The per-vertex linear mappings are the pose blendshapes $b_P : \mathcal{I} \mapsto \mathbb{R}^{3 \times |\boldsymbol{\theta}|}$, shape blendshapes $b_S : \mathcal{I} \mapsto \mathbb{R}^{3 \times |\boldsymbol{\beta}|}$, applied to a canonical template, followed by linear blend skinning with parameters $w_i : \mathcal{I} \mapsto \mathbb{R}^K$. The $i$-th vertex $\mathbf{v}_i$ is transformed according to

$$\mathbf{v}'_i = \sum_{k=1}^{K} w(\mathbf{v}_i)_k G_k(\boldsymbol{\theta}, \boldsymbol{\beta}) \cdot (\mathbf{v}_i + b_P(\mathbf{v}_i) \cdot \boldsymbol{\theta} + b_S(\mathbf{v}_i) \cdot \boldsymbol{\beta}) \tag{3}$$

where $G_k(\boldsymbol{\theta}, \boldsymbol{\beta}) \in SE(3)$ is the $4 \times 4$ transformation matrix of part $K$, see [48]. Note that SMPL is a function defined on the vertices $\mathbf{v}_i$ of the template, while we need a continuous mapping in $\mathbb{R}^3$.
Let $\psi : \mathcal{I} \mapsto \mathcal{Y}$ be a function defined on discrete vertices $\mathbf{v}_i \in \mathcal{I}$, with co-domain $\mathcal{Y}$. The idea is to derive a function $g^\psi(\mathbf{p}) : \mathbb{R}^3 \mapsto \mathcal{Y}$ which diffuses $\psi$ to $\mathbb{R}^3$. $\psi$ can trivially be diffused to the surface by barycentric interpolation and to $\mathbb{R}^3$ using the closest surface point $\mathbf{c} = \arg\min_{\mathbf{c} \in \mathcal{M}_T} \|\mathbf{c} - \mathbf{p}\|$,

$$g^\psi(\mathbf{p}) = \alpha_1 \psi(\mathbf{v}_l) + \alpha_2 \psi(\mathbf{v}_k) + \alpha_3 \psi(\mathbf{v}_m), \tag{4}$$

where $\alpha_1, \alpha_2, \alpha_3 \in \mathbb{R}^3$ are barycentric coordinates of $\mathbf{c} \in \mathcal{M}_T$, and $\mathbf{v}_l, \mathbf{v}_k, \mathbf{v}_m \in \mathcal{I}$ are corresponding canonical vertices, see Fig. 2-A,C,D.
During learning, we need to evaluate $g^\psi(\mathbf{p})$ and compute its spatial gradient $\nabla_\mathbf{p} g^\psi(\mathbf{p})$ efficiently. Hence, we pre-compute $g^\psi(\mathbf{p})$ in a 3D grid around the surface $\mathcal{M}_T$ (we use a unit cube with 64x64x64 resolution), and use tri-linear interpolation to obtain a continuous differentiable mapping in regions near the surface $\mathcal{H} \subset \mathbb{R}^3$.

**LoopReg.** The aforementioned representation allows the formulation of correspondence prediction in a self-supervised loop, as a composition of the backward map $f_\phi$ and the forward map $g^\psi$. The backward map $f_\phi(\mathbf{s}; \mathcal{S}) : \mathbb{R}^3 \times \mathcal{S} \mapsto \mathcal{H} \subset \mathbb{R}^3$ (implemented as a deep neural network) transforms every scan point $\mathbf{s}$ to the corresponding canonical point $\mathbf{p}$ in the unshaped and unposed space $\mathcal{H} \subset \mathbb{R}^3$. For clarity, we simply use $f_\phi(\mathbf{s})$ for the network prediction.
The forward map $g^\psi(\mathbf{p})$ is the diffused SMPL function by setting $\psi = M$ (body model). Specifically,

we diffuse the pose and shape blend-shapes, and skinning weights to the 3D region $\mathcal{H}$, obtaining functions $g^{b_P}$, $g^{b_S}$, $g^w$. We also obtain the function $g^I$ which maps every point $\mathbf{p} \in \mathcal{H}$ to its closest surface point $\mathbf{c} \in \mathcal{M}_T$. The diffused SMPL function for $\mathbf{x} = \{\boldsymbol{\theta}, \boldsymbol{\beta}, \mathbf{D}\} \in \mathcal{X}'$ is obtained as

$$g^M : \mathcal{H} \times \mathcal{X}' \mapsto \mathbb{R}^3, \quad g^M(\mathbf{p}, \boldsymbol{\theta}, \boldsymbol{\beta}, \mathbf{D}) = \sum_{k=1}^{K} g_k^w(\mathbf{p}) G_k(\boldsymbol{\theta}, \boldsymbol{\beta})(g^I(\mathbf{p}) + g^{b_P}(\mathbf{p})\boldsymbol{\theta} + g^{b_S}(\mathbf{p})\boldsymbol{\beta}), \quad (5)$$

which is continuous and differentiable. This enables re-formulating Eq. (2) as a differentiable loss

$$L_{\text{self}}(\phi, \mathcal{X}) = \sum_{j=1}^{N_u} \sum_{\mathbf{s}_i \in \mathcal{S}_j} \text{dist}(\mathbf{s}_i, g_{\mathbf{x}_j}^M(f_\phi(\mathbf{s}_i))) + \lambda \cdot d(f_\phi(\mathbf{s}_i)), \quad (6)$$

where $f_\phi(\mathbf{s}) = \mathbf{p}$ is the corresponding point predicted by the network, and we used the notation $g_{\mathbf{x}_j}^M(\mathbf{p}) = g^M(\mathbf{p}_j, \boldsymbol{\theta}_j, \boldsymbol{\beta}_j, \mathbf{D}_j)$ for clarity, and $d(\cdot)$ is the distance transform. Network predictions which deviate from the surface are penalized with the term $\lambda \cdot d(f_\phi(\mathbf{s}_i))$ following a Lagrangian formulation of constraints. Note that this term is important in forcing predicted correspondences to be close to the template surface, as gradient updates far away from the surface may not be well behaved. Notice that $\nabla d(\mathbf{p}_j)$ points towards the closest surface point (see Fig. 3 B.), and along this direction $g_{\mathbf{x}_j}^M(\mathbf{p}_j)$ is constant, and hence $\nabla d(\mathbf{p}_j) \perp \nabla g_{\mathbf{x}_j}^M(\mathbf{p}_j)$; the terms are complementary and do not compete against each other. Unfortunately, directly minimizing Eq. 6 over network $f_\phi$ and instance specific parameters $\mathcal{X}$ is not feasible. The initial correspondences predicted by the network are random, which leads to unstable model fitting and a non-convergent process.

**Semi-supervised learning.** We propose the following semi-supervised learning strategy, where we warm-start the process using a small labeled dataset $\{\mathcal{S}_j, \mathcal{M}(\mathbf{x}_j)\}_{j=1}^{N_s}$, with $\mathcal{M}(\mathbf{x}_j)$ denoting the registered SMPL surface to the scan, and subsequently train with a larger un-labeled corpus of scans $\{\mathcal{S}_j\}_{j=1}^{N_u}$. Learning entails minimizing the following three losses over correspondence-network parameters $\phi$ and instance specific model parameters $\mathcal{X} = \{\mathbf{x}_j\}_{j=1}^{N_s}$:

$$L = L_{\text{unsup}} + L_{\text{sup}} + L_{\text{reg}}. \quad (7)$$

The unsupervised loss consists of a data-to-model term $L_{\text{d}\mapsto\text{m}} = \text{dist}(\boldsymbol{s}, \mathcal{M}(\mathbf{x}))$ and the self-supervised loss $L_{\text{self}}$, in Eq. 6

$$L_{\text{unsup}}(\phi, \mathcal{X}) = \sum_{j=1}^{N_u} \sum_{\mathbf{s}_i \in \mathcal{S}_j} \text{dist}(\boldsymbol{s}_i, \mathcal{M}(\mathbf{x}_j)) + \text{dist}(\mathbf{s}_i, g_{\mathbf{x}_j}^M(f_\phi(\mathbf{s}_i))) + \lambda \cdot d(f_\phi(\mathbf{s}_i)), \quad (8)$$

where $\mathcal{M}(\mathbf{x}_j)$ is the SMPL mesh deformed by the *unknown* parameters $\mathbf{x}_j$, and $\text{dist}(\boldsymbol{s}, \mathcal{M}(\mathbf{x}))$ is a differentiable point-to-surface distance. The term $L_{\text{d}\mapsto\text{m}}$ pulls the deformed model $\mathcal{M}(\mathbf{x})$ to the data, which in turn makes learning the correspondence predictor $f_\phi$ more stable.

The supervised loss, $L_{\text{sup}}$, minimises the $L_2$ distance between network-predicted correspondences and ground truth ones $\hat{\mathbf{c}}_{ji}$ obtained from $\mathcal{M}(\mathbf{x}_j)$

$$L_{\text{sup}}(\phi) = \sum_{j=1}^{N_s} \sum_{\mathbf{s}_i \in \mathcal{S}_j} \|f_\phi(\mathbf{s}_i, \mathcal{S}_j) - \hat{\mathbf{c}}_{ji}\|_2. \quad (9)$$

The regularisation term $L_{\text{reg}}$ consists of priors on the SMPL shape and pose ($L_\theta$) parameters.

$$L_{\text{reg}}(\phi, \mathcal{X}) = \sum_{j=1}^{N_u} L_\theta(\boldsymbol{\theta}_j) + \|\boldsymbol{\beta}_j\|_2 \quad (10)$$

Our experiments show that with good initialization, our approach can be trained self-supervised, using only $L_{\text{unsup}}$ and $L_{\text{reg}}$.

**CorrNet: Predicting scan to model correspondences** The aforementioned formulation is continuous and differentiable with respect to instance specific parameters $\mathbf{x}_j = \{\boldsymbol{\theta}_j, \boldsymbol{\beta}_j, \mathbf{D}_j\}$ as well as global network parameters $\phi$. In this section, we describe our correspondence prediction network, CorrNet. CorrNet $f_\phi(\cdot)$, is designed with a PointNet++ [62] backbone and regresses for each input

| Registration Errors | vertex-to-vertex (cm) | | surface-to-surface (mm) | |
|---|---|---|---|---|
| Dataset | Our | FAUST [18] | Our | FAUST [18] |
| (a) Optimization [84, 42, 8] | 16.5 | 13.5 | 12.6 | 3.8 |
| (b) 3D-CODED single init. | 22.3 | 21.9 | 8.7 | 10.6 |
| (c) 3D-CODED [32] | 2.0 | 3.1 | 2.4 | 2.6 |
| (d) Ours | **1.4** | **2.2** | **1.0** | **2.4** |

Table 1: We compare registration performance of our approach against instance specific optimization (a) [84, 42, 8] (without pre-computed joints and manual selection) and learning based [32] approach. Note that 3D-CODED requires multiple initializations (c) to find best global orientation, without which the approach easily gets stuck in a local minima (b). Since [32] freely deforms a template, for a fair comparison, we use SMPL+D model for our and [42, 8, 84] approaches, even though data contains undressed scans.

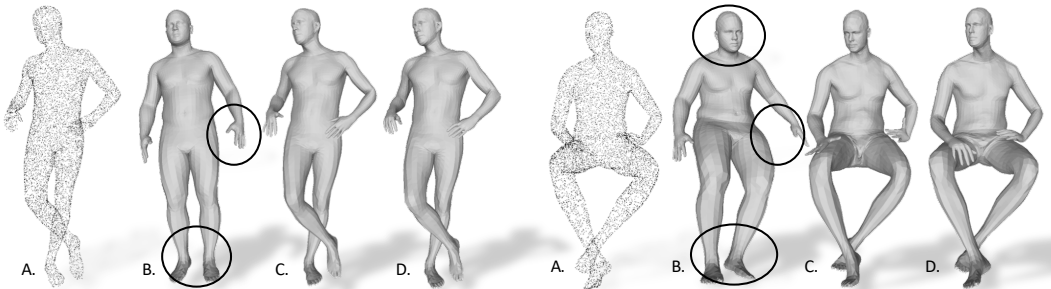

Figure 4: Comparison with existing scan registration approaches [42, 8]. We show A) input point cloud, B) registration using [42, 8] without pre-computed joints/ landmarks. C) Our registration. D) GT registration using [42, 8] + precomputed 3D joints + facial landmarks + manual selection. It can be seen that B) makes significant errors as compared to our approach C).

scan point $\mathbf{s}_i$, its correspondence $\mathbf{p}_i \in \mathcal{H}$. We observe that the correspondence mapping is discontinuous when the scan has self-occlusion and contact. For example, when the hand touches the hip, nearby scan points need to be mapped to distant $\mathbf{p} = (x, y, z)$ coordinates, which is difficult. Inspired by [63], we first predict a body part label (we use $N = 14$ pre-defined parts on the SMPL mesh) for each scan point and subsequently regress the continuous $x, y, z$-correspondence only within that part. Similar to ensemble learning approaches we use a weighted sum of part-specific classifiers to regress correspondences. This way, our formulation is differentiable with respect to part classification, which would not be possible if we directly used $\arg\max$ for hard part assignment:

$$f_\phi(\mathbf{s}) = \mathbf{p} = \sum_{k=1}^{N_{\text{parts}}} f_{\phi,k}^{\text{class}}(\mathbf{s}, \mathcal{S}) f_{\phi,k}^{\text{reg}}(\mathbf{s}, \mathcal{S}), \tag{11}$$

where $f_\phi^{\text{class}} : \mathbb{R}^3 \times \mathcal{S} \mapsto \mathbb{R}^{N_{\text{parts}}}$ is the (soft) part-classification branch of the CorrNet and $f_{\phi,k}^{\text{reg}} : \mathbb{R}^3 \times \mathcal{S} \mapsto \mathcal{H} \subset \mathbb{R}^3$ is the part specific regressor.

## 4 Experiments

In this section, we evaluate and show that our approach outperforms existing scan registration approaches. Moreover, our approach seamlessly generalizes across undressed (comparatively easier) and fully clothed (significantly more challenging) scans in complex poses. We show that our approach can be trained with self-supervision (with supervised warm-start) and performance improves noticeably as more and more raw scans are made available to our method.

### 4.1 Dataset

We use 3D scans of humans from RenderPeople, AXYZ and Twindom [2, 3, 4]. To obtain reference registrations for evaluation, we fit the SMPL model to scans using [8, 42] with pre-computed 3D joints lifted from 2D detections, facial landmarks and manually select the good fits. The input to our method are point clouds, which we extract from SMPL fits for undressed humans, and from the raw

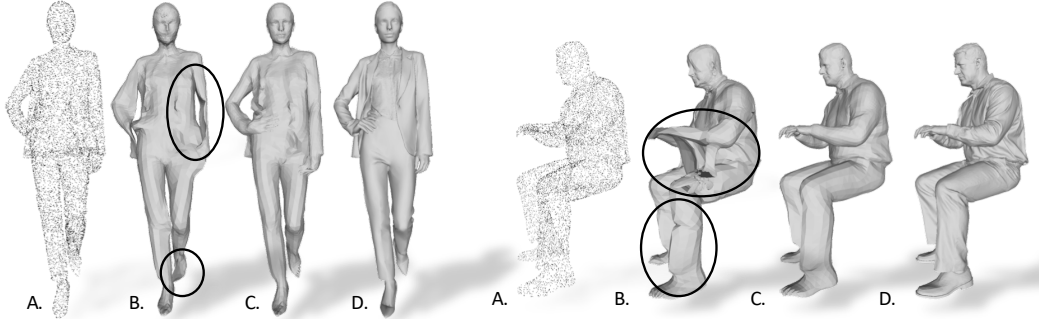

Figure 5: Comparison with existing scan registration approaches [42, 8]. We show A) input point cloud, B) registration using [42, 8] without pre-computed joints/ landmarks. C) Our registration and D) GT scan. It can be seen that B) makes significant errors as compared to our approach C).

| Unsupervised % | 0% | 10% | 25% | 50% | 75% | 100% |
|---|---|---|---|---|---|---|
| (a) v2v (cm) | 9.3 | 8.4 | 6.3 | 4.1 | 2.7 | 1.5 |
| (b) s2s (mm) | 6.8 | 6.6 | 6.2 | 5.5 | 5.1 | 4.2 |

Table 2: Performance of the proposed approach increases as we add more unsupervised data for training. Here 100% corresponds to 2631 scans. Out of the 2631 scans 1000 were also used for supervised warm-start. We report vertex-to-vertex (v2v) and bi-directional surface-to-surface (s2s) errors and clearly show that adding more unsupervised data improves registration performance, specially for the more demanding v2v metric.

scans for clothed humans. We divide the SMPL fits in a supervised (1000 scans), unsupervised (1631 scans) and testing set (290 scans). We perform additional experiments on Faust [18] which contains around 100 scans of undressed people and corresponding GT SMPL registrations.

## 4.2 Comparison with existing instance specific optimization based approaches

**Registering undressed scans.** One of the key strengths of our approach is the ability to register 3D scans without additional information such as pre-computed joint/ landmarks and manual intervention. In Fig. 4 we show that existing state of the art registration approaches [84, 42, 8] cannot perform accurate registration without these pre-processing steps.

**Registering dressed scans.** In Fig. 5 we show results with dressed scans and report an avg. surface-to-surface error after fitting the SMPL+D model of 2.2mm (Ours) vs 2.9mm ([42, 8, 84]). It can be clearly seen that without pre-computed joints, prior approaches perform quite poorly, especially for complex poses. We quantitatively corroborate the same (for undressed scans) in Table 1.

## 4.3 Comparison with existing learning based approach

Conceptually, we found the work by Thibault *et al.* [32] very related to our work, even though they require supervised training. For a fair comparison, we retrain their networks on our dataset and compare the registration error against our approach. Quantitative results in Table 1 clearly demonstrate the better performance of our approach. We also found that [32] is susceptible to bad initialization and hence requires multiple (global rotation) intializations for ideal performance. We report these numbers also in Table 1. Originally, the method in [32] was trained on SURREAL [78] with augmentation yielding a significantly larger dataset than ours. It is possible that the method of [32] requires a lot of data to perform well. Importantly, the supervised approach [32] does not deal with dressed humans whereas our approach works well for both dressed and undressed scans.

## 4.4 Correspondence prediction

Though our work does not directly predict correspondences between two shapes we can still register the two shapes with a common template. This allows us to establish correspondences between the shapes. We compare the performance of our approach on the correspondence prediction task on FAUST [18]. We report the results in Table 4. For competing approaches we take the numbers from the corresponding papers. See supplementary for further discussion.

| Supervised % | 0% | 10% | 25% | 50% | 75% | 100% |
|---|---|---|---|---|---|---|
| (a) v2v (cm) | 16.0 | 12.4 | 11.5 | 8.3 | 5.4 | 1.5 |
| (b) s2s (mm) | 13 | 7.8 | 8.5 | 7.7 | 6.4 | 4.2 |

Table 3: We study the effect of reducing the amount of available supervised data. Here 100% corresponds to 1000 scans used for supervised warm-start. We use additional 1631 scans for unsupervised training. We report vertex-to-vertex (v2v) and bi-directional surface-to-surface (s2s) errors.

| Method | Inter-class AE (cm) | Intra-class AE (cm) |
|---|---|---|
| FMNet [52] | 4.83 | 2.44 |
| FARM [49] | 4.12 | 2.81 |
| LBS-AE [44] | 4.08 | 2.16 |
| 3D-CODED [32] | 2.87 | 1.98 |
| **Ours** | **2.66** | **1.34** |

Table 4: Comparison with existing correspondence prediction approaches. Our registration method clearly outperforms the existing supervised [52, 32] and unsupervised [44, 49] approaches.

### 4.5 Importance of our semi-supervised training

**Adding more unsupervised data improves registration.** An advantage of our approach over existing approaches is the self-supervised training. We use a small amount of supervised data to warm start our method and subsequently the performance can be improved by throwing in raw scans (see Table 2). It can be clearly seen that performance improves significantly as more and more unlabeled scans are provided to our method.

**Importance of supervised warm-start.** Good initialization is important for our network before it can adequately train using self-supervised data. In Table 3 we demonstrate the importance of good initialization using a supervised warm start. Note that we use only 1000 scans for supervised warm start where as methods such as 3D-CODED [32] require an order of magnitude more supervised data for optimal performance.

## 5 Conclusions

We propose *LoopReg*, a novel approach to semi-supervised scan registration. Unlike previous work, our formulation is end-to-end differentiable with respect to both the model parameters and a learned correspondence function. While most of the current state of the art registration is based on instance specific optimization, our method can leverage information across a corpus of unlabeled scans. Experiments show that our formulation outperforms existing optimization and learning-based, approaches. Moreover, unlike prior work, we do not rely on additional information such as precomputed 3D joints or landmarks for each input, although these could be integrated in our formulation, as additional objectives, to improve results. Our second key contribution is representing parametric model as zero levelset of a distance field which allows us to diffuse the model function from the model surface to entire $\mathbb{R}^3$. Our formulation based on this representation can be useful for a wide range of methods as it removes the pre-requisite of computing a 2D surface parameterization. In contrast, we make predictions in the unconstrained $\mathbb{R}^3$ and subsequently map them to the model surface while still preserving differentiability. This makes our formulation easy to use, and potentially relevant for future work in learning based model fitting and correspondence prediction.

## Acknowledgments and Disclosure of Funding

Special thanks to RVH team members [6], and reviewers, their feedback helped improve the quality of the manuscript significantly. We thank Twindom [4] for providing data for this project. This work is funded by the Deutsche Forschungsgemeinschaft (DFG, German Research Foundation) - 409792180 (Emmy Noether Programme, project: Real Virtual Humans) and Google Faculty Research Award.

## Broader Impact

Our work focuses on registering 3D scans with a controllable parametric model. Scan registration is a basic pre-requisite for many computer graphics and computer vision applications. Current approaches require manual intervention to accurately register scans. Our method could alleviate this restriction allowing for 3D data processing in aggregate. This line of work is especially important for applications such as animation, AR, VR, or gaming.

One challenge of similar work (including ours) in human-centric 3D vision is 'limited testing'. Given that 3D data is still limited compared to 2d images, extensive testing and demonstration of generalization is difficult. This makes systems brittle to out-of-sample inputs (e.g., in our case, rare human poses). This bottleneck needs to be overcome before this and similar work can find application in fields where reliability is important.

With advancements in deep learning, a lot of current work requires collecting, processing and storing personal 3D human data. At this point in time, the awareness amongst general population regarding the negative potential of using this data is still relatively low. This could lead to privacy challenges without subjects even understanding the consequences. Processing data in aggregate as pursued here, as well as other forms of federated learning could offer convenient usability-privacy trade-offs, moving forward.

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
