[Supplementary Material]

# *LoopReg*: Supplementary Material

In this supplementary material we first present a legend of the notations used in the work to assist in reading the main paper. We then show additional results for the task of correspondence prediction and more qualitative results of scan registration using our approach.

## 1 Legend for Notations

We realise that the paper is a bit intensive in terms of notations. For improved readability, we present the definition of key symbols used in the main paper in Table 1.

| Symbol | | Meaning |
|---|---|---|
| $\mathcal{H}$ | : | Subspace of $\mathbb{R}^3$. In this work it is a unit cube around origin. |
| $\mathcal{S}$ | : | 3D Scan. |
| $\mathbf{s}/\mathbf{s}_i$ | : | Points on scan surface. |
| $M$ | : | Parametric human model such as SMPL defined on model vertices. |
| $M'$ | : | Human model function extended to non-vertex points on the surface of the model using barycentric interpolation. |
| $\mathbf{v}_i \in \mathcal{I}$ | : | Vertices on the human model $M$. |
| $\mathbf{x} \in \mathcal{X}'$ | : | Parameters for the human model $M$. |
| $\mathcal{M}_T$ | : | Surface of human model in canonical pose and shape. |
| $\boldsymbol{\theta}, \boldsymbol{\beta}, \mathbf{D}$ | : | SMPL parameters for pose, shape and non-rigid deformations. |
| $\mathbf{c}/\mathbf{c}_i$ | : | Correspondences to the human model $M$ in canonical pose and shape. |
| $\mathrm{dist}(\cdot, \cdot)$ | : | a distance metric in $\mathbb{R}^3$. |
| $f_\phi(\cdot)$ | : | CorrNet, network for correspondence prediction. |
| $N_u$ | : | Number of unlabeled scans. |
| $\mathbf{p}$ | : | A point in $\mathcal{H} \subset \mathbb{R}^3$. |
| $d(\cdot)$ | : | Distance transform of $\mathcal{M}_T$. |
| $b_S, b_P, w$ | : | shape blend-shape, pose blend-shape and skinning weights for SMPL model. |
| $\mathbf{v}_i'$ | : | Transformed vertex $\mathbf{v}_i$ after applying the pose and shape dependent deformation, non-rigid deformation and articulation according to the human model $M$. |
| $G_k(\boldsymbol{\theta}, \boldsymbol{\beta})$ | : | SMPL skeletal transformation matrix. |
| $K$ | : | Number of joints in the SMPL model (K=24). |
| $\psi$ | : | Arbitrary (differentiable) function defined on discrete mesh vertices $\mathbf{v}_i \in \mathcal{I}$. |
| $\mathcal{Y}$ | : | Co-domain of $\psi$. |
| $g^\psi$ | : | Function that diffuses $\psi$ from the surface of the human model $M$ to $\mathcal{H} \subset \mathbb{R}^3$. |
| $g^M$ | : | Diffused human model from surface to $\mathcal{H}$. |
| $g^I, g^w, g^{b_P}, g^{b_S}$ | : | Components of the diffused SMPL function, namely closet point function, skinning weight function, pose blend-shape and shape blend-shape function. |
| $g_{\mathbf{x}}^M$ | : | Diffused human model for the input parameter $\mathbf{x}$. |
| $\nabla$ | : | Gradient operator. |
| $L_{\mathrm{self}}(\phi, \mathcal{X})$ | : | Self-supervised loss over correspondence network parameters $\phi$ and instance specific human model parameters $\mathcal{X}$. |
| $L_{\mathrm{d}\rightarrow\mathrm{m}}(\mathbf{s}, \mathcal{M})$ | : | Distance between the point $\mathbf{s}$ and mesh $\mathcal{M}$. Unsupervised loss. |
| $L_{\mathrm{unsup}}$ | : | Unsupervised loss. $L_{\mathrm{unsup}} = L_{\mathrm{self}} + L_{\mathrm{d}\rightarrow\mathrm{m}}$. |
| $L_{\mathrm{sup}}$ | : | Supervised loss on predicted correspondences. |
| $L_{\mathrm{reg}}$ | : | Losses based on priors on human model parameters. |
| $f_\phi^{\mathrm{class}}$ | : | Part classification branch of CorrNet $f_\phi$. |
| $f_{\phi,k}^{\mathrm{reg}}$ | : | Correspondence regressor corresponding to part $k$. Sub-part of CorrNet. |

Table 1: Key notations used in the paper.

| Method | Inter-class AE (cm) | Intra-class AE (cm) |
|---|---|---|
| FMNet [7] | 4.83 | 2.44 |
| FARM [6] | 4.12 | 2.81 |
| LBS-AE [5] | 4.08 | 2.16 |
| 3D-CODED [3] | 2.87 | 1.98 |
| **Ours** | **2.66** | **1.34** |

Table 2: Comparison with existing correspondence prediction approaches. Our registration method clearly outperforms the existing supervised [7, 3] and unsupervised [5, 6] approaches even though it is not directly trained for establishing correspondences across shapes.

## 2 Results: Correspondence prediction

Establishing correspondences across 3D shapes is a challenging problem in computer graphics and vision community. Though our work does not directly predict correspondences between two shapes we can still register the two shapes with a common template. This allows us to establish correspondences between the shapes. We compare the performance of our approach on the correspondence prediction task on FAUST [2]. The FAUST test set contains 200 scans of undressed people in challenging poses and the scans themselves are noisy. The evaluation metric is based on the geodesic distance between the predicted correspondence and the GT correspondence. This metric heavily penalises the errors made due to self contacts on the body. In practice it can be seen that the overall distribution of errors is dominated by the contact errors making the evaluation less than ideal. Nonetheless we report the results as per the protocol in Table 2. For competing approaches we take the numbers from the corresponding papers. It can be clearly seen that our model trained primarily with self-supervision performs better than the competing approaches. Also note that none of these other approaches generalises to dressed humans where as in Fig. 3 we show that our method can predict correspondences for both dressed and undressed scans.

## 3 More qualitative results

We show additional qualitative results for undressed scan registration in Fig. 1 and for dressed scans in Fig. 2. It can be seen that our approach can produce high quality registrations for both undressed and dressed scans in complex poses. Our approach predicts continuous correspondences from the scan to the canonical human template. We visualise these correspondences in Fig. 3 and show that we can accurately predict correspondences for both undressed and dressed scans.

## 4 Limitations and Future Work

Our formulation allows us to jointly differentiate through the correspondences and the instance specific human model parameters. This allows us to create a self-supervised loop for registration. But in practice we find that in order for this loop to not get stuck in a local minima, it is important that correspondences are initialized well. So even though our formulation does not require labeled data, in practice we find that a supervised warm-start with a small amount of data is important for subsequent self-supervised training.

As shown in our results, our approach performs significantly better than other competing approaches both qualitatively and quantitatively. We still find that our registration is not as high quality as [4, 1] when they have access to precomputed 3D joints, facial landmarks and manual intervention (Note that our approach does not require this information). This is not necessarily a limitation as these additional cues can be integrated with our approach as well. We leave this as a potential future work.