[Reviews · NeurIPS 2020]

Review 1

Summary and Contributions: This paper proposes an approach to register 3D scans to a 3D human parametric model. Interestingly, even though the parametric model represents a naked body, the approach works both for scans of naked and dressed people (although without wearing loose clothes). I consider the paper makes two main contributions: 1) an implicit representation of a human model template using a zero levelset of a signed distance field. While this is not novel, it was not used for this specific problem; 2) a differentiable cycle optimization loss that maps from the input scan to a canonical template and back to the input scan. This allows a self-supervised learning without the need of labeled scans. Note, however that this process requires a ‘warm up’ with fully supervised data. The approach compares favorably to state-of-the-art.

Strengths: -The whole methodology builds upon a solid mathematical framework. -The implicit surface representation is a nice trick to directly estimate point-to-surface distance. -The scan-model-scan mapping that allows training without the need of labelled data is an interesting contribution. -The solid methodology is curated with different implementation tricks (e.g. combination of supervised/non-supervised stages, regularization loss, etc.) -The authors mention they will distribute the code. This is also a positive point.

Weaknesses: -The comparison with current SOA [Groueix et al. ECCV 2018] should be more exhaustive in order to be fully convincing. For instance, I miss an experiment showing the robustness of the proposed approach to noise. This should be evaluated on the examples of Figure 3, with the naked body, in which there exist a perfect match between the scan and the 3D mesh. [Groueix et al. ECCV 2018] was shown to be robust to perturbations. It would be interesting to see if the current approach is also robust. I presume it will be, as it works for scans of dressed people, but a direct comparison to [Groueix et al. ECCV 2018] in this context would be more convincing. -The authors claim their approach is self-supervised, while it needs an initial warm-up with fully supervised data. The size number of supervised samples is almost as large as the non-labelled set (1000 vs 1631). Without using a 100% of the fully supervised data, the results would not improve the previous SOA method [Groueix et al. ECCV 2018]. Therefore, I believe the authors should tone down their self-supervision claim.

Correctness: The claims are very clear and the method is simple and correct.

Clarity: The paper is very well written.

Relation to Prior Work: Reference to prior work is excellent.

Reproducibility: Yes

Additional Feedback: =========== COMMENT AFTER REBUTTAL PHASE======= The authors have partially addressed the issues I raised in the review. In particular I’d have expected an evaluation of the proposed approach to larger amounts of noise than those in the FAUST dataset. [Groueix et al. ECCV 2018] provide evaluation on perturbed versions of the TOSCA dataset with quite large holes and amounts of noise. In any event, I still believe the paper deserves being accepted.


Review 2

Summary and Contributions: This paper proposes a self-supervised approach for jointly learning to register acquired 3D scans (point clouds) of human subjects to statistical parametric shape models (SMPL-D). The authors propose to construct a differentiable end-to-end trainable loop to find correspondences to the canonical model for the scan points and also to find the deformation parameters of the model to match the scan. The key ideas presented to solve this problem are to represent the parametric model's surface in 3D as a signed distance field instead of a UV parameterization and, secondly, to diffuse the functions on the model's surface to R^3 around the surface. The authors show comparisons to several competing approaches and outperform all of them.

Strengths: Labeling 3D scans with landmarks is currently a necessary, but highly labor-intensive process to enable accurate registration of 3D scans to a standard canonical templates. This is a step required to convert scans to a standard topology. Creating methodologies to automate this process as much as possible, as this work proposes, can be very impactful in allowing us to build richer statistical 3D models more quickly and with many more scans. So this work is well motived and has the potential to have a very positive impact on the field along with the availability of its code and pertained models. The proposed method proposes several innovative ideas to solve the problem including using a signed distance field as the parametric representation, along with diffusion of the parametric model's parameters to a 3D volume around the surface to enable an end-to-end differentiable formulation. These ideas are quite general and are not limited to fitting of body scans only. They could potentially be adopted in other 3D-related problems as well. The results presented are quite impressive and even with less supervision the proposed approach outperforms the competing approach of [26], which is fully-supervised. Furthermore, it is applicable to both clothed and unclothed humans. The paper is very well written and is a pleasure to read. The method is conceptually sound.

Weaknesses: This is not a weakness per-say, but a suggestion to make the paper stronger. In juxtaposition to the existing work the authors present the argument several times that using a UV parameterization is inherently inferior to 3D representations, as it requires seam-cuts and results in distortion of highly curved regions, etc. While this is conceptually correct and true, it would have made the paper stronger if the authors had somehow demonstrated this to be true empirically as well for their problem. For example, perhaps via a simpler problem -- maybe for the fully-supervised case or for the case when the entire pipeline is not necessarily end-to-end differentiable, but a combination of a landmarks/correspondence estimation + a traditional optimization approach. It would be interesting to see if the signed distance representation to predict correspondences with a CNN along with its Lagrangian loss formulation to encourage points to lie on the surface improves the accuracy of correspond prediction by itself and if so by how much versus an approach that learns to map scan points to the UV space instead. The second comment is with regards to the clarity of the section titled "Diffusing the human model function to R^3". I found this section in general and especially line 176 hard to conceptually comprehend, requiring several re-reads to correctly conceptualize. This is one of the central ideas of the paper and hence fundamental to its understanding. It would be super helpful if the authors could illustrate the idea of diffusing the function of the surface a bit more concretely, perhaps via a simpler toy example. Lastly, it would also be helpful to visualize some failure cases of the current approach. Also, what is its runtime?

Correctness: Yes.

Clarity: Yes.

Relation to Prior Work: Yes.

Reproducibility: Yes

Additional Feedback: Line 65: "function the" should be "function into the" Line 115: "This prone" should be "This is prone" Post Rebuttal: I have read all the reviews and the author's response. I agree with the other reviewers that because of the warm-start required by the proposed method with 1000 supervised images, it is not completely fair to call the method a "self-supervised". The authors have agreed to tone down that claim. I have lowered my score to 8 from 9 for that reason. Other than that I think this a solid paper with many innovative ideas, which will positively impact research in 3D reconstruction and model-fitting.


Review 3

Summary and Contributions: This paper presents an end-to-end learning pipeline for registering parametric human body models to 3D scans. The key idea is to derive a differentiable way to diffuse the SMPL model onto the 3D space using an implicit surface representation and barycentric interpolation, which allows for self-supervised training via a closed loop from scan point to canonical surface then back to 3D. Using a small dataset with labeled correspondences for pretraining and the proposed correspondence cycle constraint, the model is able to achieve accurate, stable registration results without requiring keypoint supervision, outperforming SOTA methods using full supervision. The authors also demonstrate such a self-supervised training scheme can improve the performance on top of supervised baselines by using more unlabeled data.

Strengths: The idea constructing a closed loop for leveraging self-supervision makes a lot sense, and is implemented in a smart way in this paper. In order to close this loop between 3D point and parametric model, the proposed method first constructs a SDF as an implicit representation of the surface of a canonical SMPL model. It then diffuses the parametric SMPL surface onto the 3D space, such that the correspondences can be easily established from 3D to SMPL surface and back to 3D in a end-to-end differentiable pipeline. The results demonstrates that with such a cycle constraint, the performance can be significantly boosted using additional unlabeled data, beating fully supervised SOTA methods. It also bypass the need for keypoints and multiple ratation initializations used in other methods. I like Table 2 & 3, in which they show the performance when increasing unlabeled data and decreasing labeled data, which is insightful. In general, I think this cycle idea is definitely useful and also the way they reparametrize the surface of the SMPL model are quite inteteresting and inspiring.

Weaknesses: I do not have anything major to complain, but I find it a bit disappointing to realize that the proposed method also requires a "small" set of labeled data for pretraining, which is also not that small (1000 labeled vs 1631 unlabeld ones), despite the repeated claims of being "self-supervised" and the complaints about previous methods requiring supervision. I understand the point is to show that self-supervision using unlabeled data on top of supervised baseline improves the performance, but I think the authors should adjust the tone in the text.

Correctness: Yes.

Clarity: Yes, although it takes some time to digest. I think the link between the implicit surface representation and the diffusion part is not obvious and was rather confused during the first read. Fig 2 does not help much, and is also confusing itself. I appreciate efforts in keeping the mathematical notations rigorous.

Relation to Prior Work: Yes.

Reproducibility: Yes

Additional Feedback: Fig 1 & 2 somhow do not load in my Acrobat, but works in browsers. ----- post rebuttal ----- I think overall this is a solid submission and will remain positive. The rebuttal has addressed most of the concerns. All reviewers have pointed out that the self-supervised learning is an overclaim given that the model requires labeled data for pretraining. I encourage the authors adjust the writing accordingly.


Review 4

Summary and Contributions: The paper proposed a semi-supervised method to fit 3D clothed/unclothed human scans in a cycle-loop optimization manner. Different from classical fitting algorithm which requires pre-labeled 3D landmarks, LoopReg is an end-to-end learning framework to register a corpus of scans to a human template model, like SMPL. Its main contributions lie in the cycle-loop learning strategy and the differentiable feature, which comes from defining the canonical surface implicitly and diffusing the human model to the 3D domain. The authors also provide some experimental results to show that this learning strategy can guarantee better registration accuracy. With more unlabeled human scans, the accuracy will be further boosted.

Strengths: 1. Diffusing the human model function makes the implicit surface representation possible, this is a useful trick to make the algorithm to be differentiable. 2. The results look reasonably good, and the comparison shows that this cycle-supervision manner boost the correspondence accuracy a lot. 3. Table 2 shows that the semi-supervision strategy will get better results given more unlabeled data, showing strong potential to replace fully-supervised method in this task.

Weaknesses: 1. The self-supervision part is over-claimed. In fact, this work uses supervised learning to train the correspondence predictor for warm-start good initialization. It is more like a semi-supervised method. 2. Line 33-34 is a claim without any real support. Why predict the latent parameters from 3D point cloud is difficult? 3. The comparison in Fig 3 is not fair, the use of 3D landmarks is for good initialization, the proposed method also needs a good initialization [Line: 218-219], if the purpose is to show the B results, it is better to show the LoopReg results without the supervision loss used for warm-start initialization.

Correctness: The self-supervision part is over-claimed. The empirical methodology is correct.

Clarity: No, this paper is difficult to understand, especially the mathematic part. Also the writing needs some careful proofreading.

Relation to Prior Work: Yes.

Reproducibility: Yes

Additional Feedback: The authors' rebuttal has addressed my major concerns so I raised my score of this submission.

[Author Response · NeurIPS 2020]

We thank all the reviewers for their constructive and detailed feedback.

**Goal:** Automatically register scans of dressed and undressed 3D humans with the SMPL model.

**Key novelty/ technical contributions.** Our idea of implicitly representing SMPL as zero level set of a SDF is key
for making the correspondences differentiable and is well received by the reviewers, [R1: "implicit... nice trick"], [R2:
"several innovative ideas"], [R3: "interesting and inspiring."], [R4: "useful trick"]. Our second key contribution of
formulating registration as a closed loop (scan-model-scan) is also appreciated by the reviewers, [R1: training without...
labelled data... interesting contribution."], [R3: "closed loop... makes a lot of sense... implemented in a smart way.",
"cycle idea is definitely useful"]. The reviewers also highlight that our approach is mathematically well grounded and
sound [R2, R1: "solid mathematical framework"], and will have a positive impact outside this specific task [R2: "ideas
are quite general... not limited to fitting of body scans only... adopted in other problems"]. We will release our code.

**Results and writing.** Reviewers acknowledge the superior performance of our method as compared to the existing
methods. [R2: "results... quite impressive"], [R3: "accurate, stable registration... outperforming SOTA"]. We thank the
reviewers for acknowledging our writing effort, [R1: "very well written.", "Reference to prior work is excellent."], [R2:
"very well written... pleasure to read"], [R3: "appreciate rigorous mathematical notations"].

We address the reviewers concerns below:

**Requirement of supervised warm-start.** [R1, R2, R3, R4]
We agree with the reviewers that our method requires a super-
vised warm-start and hence is not fully self-supervised (Ab-
stract L20, Method L205, Experiments L238,267,271). We
notice that the terms *self-* and *semi-* supervised learning are a
bit confusing in the paper and we will correct this. However, to
highlight that the requirements for supervised data are not very
stringent, we show that the warm-start can be performed with
*only* un-dressed scans, and we can self-supervise our method
with dressed scans. Compared to a warm-start with labeled
dressed scans, this results in only $0.24mm$ higher error, see
Fig. 1. We note that acquiring supervised data for undressed
shapes (in the form of SMPL meshes) is relatively easy, given
existing datasets such as AMASS [Mahmood *et al.* ICCV'19],
SURREAL [Varol *et al.* CVPR'17].

Figure 1: We show registration of subjects (A) from Fig. 4 (main paper) using undressed data for warm-start. We compare undressed warm-start (B) with dressed warm-start (C) and report negligible difference.

**Robustness to noise and run-time.** [R1] To show robustness to noise, we evaluate our method on FAUST [Bogo *et*
*al.* CVPR'14] (scans containing noisy points, holes, self-intersections etc). We use our semi-supervised method (1K
supervised and ∼1.6K unsupervised) and compare performance with [26] (fully supervised, trained on ∼200K meshes)
in supp. mat. Table. 2. We outperform [26] and other competing methods. [R2] Correspondence prediction by NN
takes < 1 sec., SMPL/SMPL+D fitting depends on the complexity of the pose/clothing etc. but can be done in under 15
mins. Since this step does not take up a lot of memory, we can fit ∼200 meshes in parallel (P100 GPU 22GB memory).

**Clarifications.** [R2, R3] We will better explain the diffusion
of SMPL to $\mathbb{R}^3$. We will update Fig. 2 (main paper) and use
a toy example to better convey the diffusion process and will
update the text. [R2] We will better illustrate the problems with
UV parametrization and add failure cases to our paper. [R4] We
admit the paper is math heavy, but we found it essential to keep
the notations rigorous. We will add more text around equations
to intuitively explain the key ideas. [R4: "comparison in Fig 3"].
Our method and baselines [35, 6] require very different levels
of 'good initialization'. We require small amount of supervised
data at *training* time to warm-start our method, whereas [35, 6]
require manual selection and 3D annotations for *every test input*.
[R4: "Why predicting the latent parameters is difficult?"] This
was our initial experiment as well (predict SMPL+D params.
from scan) and as can be seen in Fig. 2, we could not get any
reasonable results. There are no existing works that do this. We can add these results to supp. mat.

Figure 2: Given a dressed subject (A) we show that directly predicting SMPL+D parameters (B) performs significantly worse than our method (C).

[Meta-Review · NeurIPS 2020]

The rebuttal addressed the main criticisms raised by the reviewers: the assumption on warm start, the robustness to noise, and the clarification of the model. The answers of the authors contributed to the discussion and the proper evaluation of this work. The terminological issue doesn't affect the final decision.